

# Reconstructing Late Holocene North Atlantic atmospheric circulation changes using functional paleoclimate networks

Jasper G. Franke[1,2], Johannes P. Werner[3], and Reik V. Donner[1]

[1]Potsdam Institute for Climate Impact Research, Telegrafenberg A31, 14473 Potsdam, Germany
[2]Department of Physics, Humboldt University, Newtonstraße 15, 12489 Berlin, Germany
[3]Bjerknes Centre for Climate Research and Department of Earth Science, University of Bergen, Postboks 7803, 5020 Bergen, Norway

*Correspondence to:* Jasper G. Franke (jasper.franke@pik-potsdam.de)

**Abstract.** Obtaining reliable reconstructions of long-term atmospheric circulation changes in the North Atlantic region presents a persistent challenge to contemporary paleoclimate research, which has been addressed by a multitude of recent studies. In order to contribute a novel methodological aspect to this active field, we apply here evolving functional network analysis, a recently developed tool for studying temporal changes of the spatial co-variability structure of the Earth's climate system, to a

set of Late Holocene paleoclimate proxy records covering the last two millenia. The emerging patterns obtained by our analysis are intimately related to long-term changes in the dominant mode of atmospheric circulation in the region, the North Atlantic Oscillation (NAO). By comparing the time-dependent inter-regional linkage structures of the obtained functional paleoclimate network representations to a recent multi-centennial NAO reconstruction, we identify strong co-variability between Southern Greenland, Svalbard and Fennoscandia as being indicative of a positive NAO phase, while connections from Greenland and

Fennoscandia to Central Europe are more pronounced during negative NAO phases. By drawing upon this correspondence, we use some key parameters of the evolving network structure to obtain a qualitative reconstruction of the NAO long-term variability over the entire Common Era (last 2000 years) using a linear regression model trained upon the existing shorter reconstruction.

## 1 Introduction

The increasing availability of high-resolution paleoclimate archives and resulting proxy records allows to not only study local climate variability before the beginning of the instrumental period, but also associated spatial structures at least at a regional level. Corresponding studies have commonly been performed using linear multivariate statistical methods like empirical orthogonal function (EOF) analysis (Gouirand et al., 2008; Mann et al., 1998) or, more recently, Bayesian Hierarchical Modeling (Luterbacher et al., 2016). However, at the conceptual level, many of the classical statistical approaches have considerable

problems in analyzing paleoclimate data. On the one hand, traditionally used estimators are often inappropriate for coping with spatially sparse and unevenly sampled time series. On the other hand, the appealing alternative of data interpolation can lead to a systematic bias and large uncertainties in the resulting reconstructions of spatial patterns of past climate variability (Rehfeld



et al., 2011). Furthermore, many previously applied methods rely on some kind of linearity and/or orthogonality assumption, which might result in some unrealistic representation of the climatic processes or phenomena under study.

Some of the aforementioned challenges can be (at least partially) addressed by the concept of functional climate networks (Tsonis et al., 2006; Donner et al., 2017), a recently developed nonlinear approach to studying climate dynamics that can also be employed for evaluating spatial co-variability among paleoclimate archives (Rehfeld et al., 2013). Here, each time series from a set of climate observations associated with different geographical locations is represented as a node of an abstract network embedded in geographical space. Pairs of such nodes are connected by links if the observed dynamics is sufficiently similar, which is referred to as *functional connectivity* to highlight that similar, mutually dependent physical processes are commonly reflected by strong co-variability. Climate networks provide an intuitive way to quantitatively account for the full complexity of co-variability and teleconnection patterns. Furthermore, instead of considering a spatially homogeneous data coverage, they simply ignore regions without data, which is particularly important in the case of sparse paleoclimate data.

Beyond the viewpoint of time-independent or average spatial co-variability patterns, evolving functional networks are constructed from the available data covering different time windows and thus allow for studying the evolution of such spatial patterns in time. While evolving functional climate networks have become a widespread tool to analyze modern climate data (Donner et al., 2017; Radebach et al., 2013), applications to paleoclimate data sets have been much less common so far (Rehfeld et al., 2013; McRobie et al., 2015; Oster and Kelley, 2016).

In this study, we highlight the potential of evolving functional paleoclimate networks for investigating climate variability during the Common Era (last 2 ka) in the European North Atlantic region. Climate dynamics within this region is of crucial importance not only at regional scales (Scaife et al., 2008; Trigo et al., 2002), but also as a pacemaker for the whole Northern Hemisphere (Delworth et al., 2016). Inter-annual to multi-decadal climate variability in the North Atlantic sector is strongly influenced by large-scale variability patterns like the Atlantic Multidecadal Circulation (AMO) (Knight et al., 2006) and North Atlantic Oscillation (NAO) (Hurrell et al., 2003).

The NAO is related with the persistent redistribution of air masses between the Arctic and the Central Atlantic (Hurrell and Deser, 2010) and is commonly defined as a pressure dipole over the North Atlantic, consisting of a predominant low pressure system over Iceland and a high pressure system close to the Azores. The strength of the gradient between both varies in time and provides a basis for the quantitative description of the NAO based on an index where high (low) values correspond to a strong (weak) gradient. This pressure gradient has severe consequences for climate variability in Europe, Greenland and North America. A positive phase of the NAO is commonly associated with more moderate temperatures and higher precipitation sums during winter in Northern Europe and the Eastern United States, whereas Greenland, Canada and Southern Europe often exhibit opposite characteristics. While the influence of the NAO phase is strongest during boreal winter, it also affects summer conditions (Ogi et al., 2003; Folland et al., 2009).

As the NAO is a key aspect of European climate variability, long-term changes in the dominant phase of this atmospheric variability mode should also be reflected in the co-variability structure of existing paleoclimate records. To this end, there are various types of archives available that could be utilized for reconstructing such changes. Specifically, most high-resolution archives in the region are sensitive tracers of inter-annual temperature variability (commonly seasonal or annual mean values)





and, hence, should have been influenced to a certain degree by the NAO. For example, ice core data from Southern Greenland mostly reflect winter temperatures and, thus, strongly follow winter NAO conditions (Appenzeller et al., 1998; Vinther et al., 2010). In turn, tree ring chronologies mainly trace summer temperatures, but can be additionally influenced by winter conditions, especially in terms of extreme precipitation anomalies (see, e.g. Lindholm et al., 2001; Linderholm and Chen, 2005;

Vaganov et al., 1999, and references therein). While these facts have been utilized to reconstruct NAO indices prior to the instrumental period (Cook et al., 1998), caution has to be taken since the corresponding relationship is non-stationary and, thus, the principle of uniformitarianism can be violated (Schmutz et al., 2000; Zorita and González-Rouco, 2002; Lehner et al., 2012). Furthermore, the actual effect may not be the same at different locations. For example, a persistent positive phase of the NAO leads to higher winter precipitation in Northern Europe, which in turn has an indirect influence on tree growth during

summer. At the same time, positive NAO phases have been connected to low precipitation and even droughts in Southern Europe. Hence, we expect tree ring records from Central and Southern Europe to be more strongly affected by negative NAO phases than by positive ones.

   Following upon these considerations, we anticipate that different types of terrestrial archives available in different parts of the North Atlantic region have in common that they all reflect the leading mode of regional climate variability at inter-annual to

multi-decadal time scales in one way or another. Hence, the main idea of this study is that by exploiting the temporary presence or absence of similar variability patterns between different paleoclimate records, especially in Southern Greenland versus the rest of the study region, one can draw conclusions about changing commonalities between the main atmospheric drivers in different sub-regions and, thus, the mean state of the atmospheric circulation in the North Atlantic region.

   The remainder of this paper is structured as follows: In Sec. 2, we describe the ensemble of paleoclimate proxy records used

in this study. Section 3 presents the methods used to construct evolving functional paleoclimate networks and derive from them a scalar index variable describing the time-dependent dominant mode of North Atlantic climate variability over the Common Era. In Sec. 4, we discuss the emerging structures and how well they reflect associated changes in the dominant NAO phase at multi-decadal to multi-centennial time scales. Our results are compared to findings from other studies in Sec. 5, including a discussion on the possibilities and possible shortcomings of our approach. The paper ends with concluding remarks in Sec. 6.

## 25   2   Data

The North Atlantic region comprises a large variety of well-studied high-resolution paleoclimate archives for the Late Holocene. Existing data sets include several ice core records from the Greenland ice shelf and Svalbard, tree ring chronologies from the Scandinavian mountains, the Alps and other mountain ranges, and varved lake sediments, especially in Southern Finland. In addition, there exist also some very long historical temperature records based upon early instrumental records (see references

in the Supplementary Material Tab. S1). Many of the available proxies are strongly correlated to seasonal or annual temperature variability and thus have been used as key input for existing regional (Luterbacher et al., 2016; PAGES 2k Consortium, 2013) and hemispheric (Ljungqvist et al., 2012; Mann et al., 2008) temperature reconstructions. While there are particularly many archives covering the last millennium, a considerably lower number spans the full Common Era at high resolution.





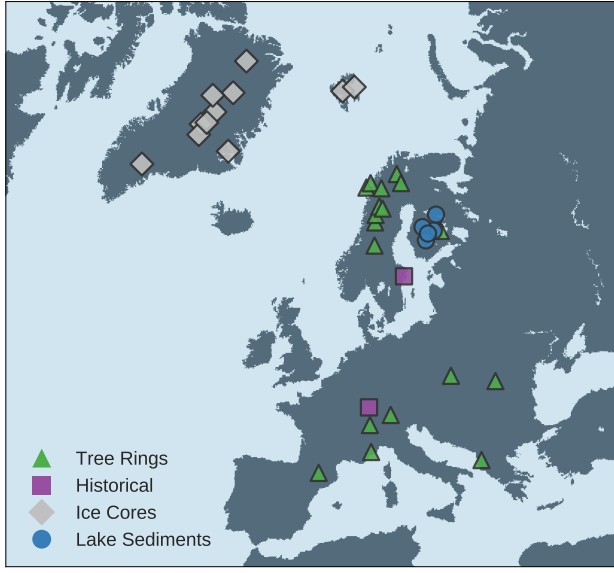

**Figure 1.** Locations and types of the paleoclimate archives used in this study. Detailed information on the individual data sets can be found in the Supplementary Material Tab. S1.

In this study, we concentrate on changes in the inter-annual to multi-decadal co-variability of temperature-sensitive proxies throughout the Common Era. Thus, we only include records into our analysis that span at least 300 years and have close to annual resolution. This leaves us with 37 time series, which are described in detail in the Supplementary Material Tab. S1 and shown in Fig. 1. Note that only 12 of these records cover the full Common Era, all of them being located in either Greenland,

Fennoscandia or the Alps.

## 3 Methods

As mentioned in the introduction, functional climate network analysis has recently become an established tool for studies on climate dynamics. Following upon the success of this approach, a few initial studies have transferred the corresponding idea to the analysis of spatial co-variability patterns among paleoclimate archives in a defined region (Rehfeld et al., 2013; McRobie

et al., 2015; Oster and Kelley, 2016). Following upon these previous works, the general workflow of functional paleoclimate network analysis is visualized in Fig. 2. Beyond the original framework, we aim here at studying the statistical interdependence structure between subsets of archives from different appropriately defined regions and relate the information inferred from this analysis to a macroscopic index tracing the dominating mode of inter-annual North Atlantic climate variability at multi-decadal time scales. Accordingly, the methodological approach followed in this work comprises the following four steps:

1. Identify an ensemble of paleoclimate proxy records that have been influenced by a common climate variable.



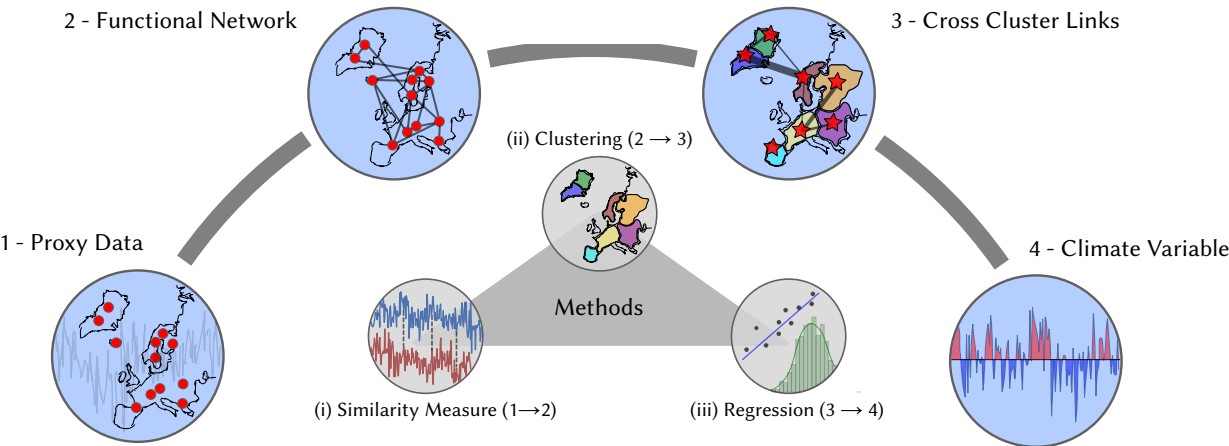

**Figure 2.** Schematic overview on the methodological approach of this study. Based upon a network of paleoclimate proxy records, we construct evolving functional networks encoding the co-variability among the different time series. Using cluster analysis, we simplify the emerging network structures and obtain quantitative measures of the inter-cluster linkages as key characteristics of the obtained networks. These variables are then related to an established long-term reconstruction of the NAO index via linear regression.

2. Construct evolving functional networks based upon this ensemble according to the mutual similarity between individual records.

3. Introduce a meaningful grouping of the records to reduce the complexity and increase robustness of the obtained information.

4. Establish a statistical relationship between the characteristics of the paleoclimate network and some climate variable or index.

While we have already discussed the first step of this workflow in the previous section, we will highlight in the following the methodological realization of the three remaining steps.

## 3.1 Functional network construction

A functional network is a graph-theoretical representation of the mutual similarity structure among a set of time series $\{x_t^i\}$ $(i=1,\ldots,N^r)$, in our case a set of $N^r=37$ paleoclimate proxy records, which are considered as nodes of the network. Here, the effective number of available records $N_t^r$ varies in time, with $\max_t N_t^r \leq N^r$, and each node is identified with the respective geographical location of the underlying paleoclimate archive. Links between different nodes are established if the



corresponding time series are significantly similar according to some corresponding measure of pair-wise statistical dependency (see below for details).

Evolving functional networks describe a time-ordered sequence of such functional networks, each being constructed from data within a given time window of length $\mathcal{W}$, which span different periods of time. Here, a time coordinate is assigned to each window according to its respective end point. Thus, any statistical measure calculated from a subset of a time series $\{x^i\}_{t^{\mathcal{W}}}$ corresponding to the time window of length $\mathcal{W}$ ending at time $t$ is associated with the set of time indices $t^{\mathcal{W}} := \{t' | 0 \le t - t' \le \mathcal{W}\}$. The window size $\mathcal{W}$ determines the temporal resolution of the analysis. The sliding windows approach allows tracing changes in the co-variability structures through time.

Each individual network is described by the $N^r \times N^r$ *adjacency matrix* $\mathbf{A}$, defining the connections between nodes,

$$A_{t^{\mathcal{W}}}^{i,j} = \begin{cases} 1 & \text{if } \{x^i\}_{t^{\mathcal{W}}} \text{ and } \{x^j\}_{t^{\mathcal{W}}} \text{ are similar} \\ 0 & \text{else.} \end{cases} \tag{1}$$

## 3.2 Similarity assessment

In general, functional networks can be constructed based upon different types of similarity measures, including classical linear approaches like the Pearson correlation, but also other measures suitable for detecting non-linear or event-based relationships, like mutual information or event synchronization (see, e.g. Rehfeld and Kurths, 2014, for a comparison of possible measures). Specifically, to determine the strength of co-variability between two paleoclimate records, a suitable similarity measure has to be able to cope with unevenly sampled and/or discontinuous time series. Here, we use a Gaussian kernel-based variant of the Pearson correlation coefficient (gXRF) (Rehfeld et al., 2011). Given two time series $\{x_i\}_{i=1}^{N^x}$ and $\{y_j\}_{j=1}^{N^y}$ with observation times $\{t_i^x\}_{i=1}^{N^x}$ and $\{t_j^y\}_{j=1}^{N^y}$, this correlation is defined as

$$\rho(x, y) = \frac{\sum_{i=1}^{N^x} \sum_{j=1}^{N^y} x_i y_j K(t_j^y - t_i^x)}{\sum_{i=1}^{N^x} \sum_{j=1}^{N^y} K(t_j^y - t_i^x)}. \tag{2}$$

Here, the kernel function $K(\cdot)$ is given by

$$K(t_j^y - t_i^x) := \frac{1}{\sqrt{2\pi h}} e^{-(t_j^y - t_i^x)^2 / 2h} \tag{3}$$

with $h = \max(\Delta t^x, \Delta t^y)/4$, where $\Delta t^{x,y}$ denote the mean sampling intervals of the corresponding time series. Specifically, as the same atmospheric driving variable can have qualitatively different effect on different proxies or at different locations, we take the absolute value of this correlation to quantify the strength of similarity.

Since our analysis makes use of different types of paleoclimate proxies, it is advisable to define similarity in a way that takes the different characteristics of the proxies into account. Time series originating from the same type of archive record climate variability in a similar fashion and thus might intrinsically exhibit stronger mutual correlations than such from different types of archives. Specifically, long-range auto-correlations and associated low-frequency variability can lead to spurious correlations, which have to be corrected for (Guez et al., 2014). To account for this problem, we apply a surrogate-based significance test



to calculate $p$-values corresponding to the probability that two records are similar just by chance, given their inherent auto-correlation structures. For this purpose, we use 1000 amplitude-adjusted Fourier transform (AAFT) surrogates (Schreiber and Schmitz, 2000), which leave the auto-correlation structure of each time series intact. Note that among the considered set of archives, all but four proxy records are complete and actually evenly distributed at annual time scale (one having lower sampling

resolution and the three others containing gaps). In this regard, using the Gaussian kernel correlation (developed for unevenly sampled data) instead of classical Pearson correlation coefficent accounts for these four records with different properties, while AAFT surrogates can still be generated with standard procedures for all records. The estimated $p$-values resulting from the surrogate ensembles provide a generally applicable measure of similarity, and two proxies are considered to be similar, if their $p$-value is below a defined threshold value $\alpha_{pr}$.

## 10  3.3  Network analysis

In the case of paleoclimate time series, there are some particular complications to be addressed in the context of functional network analysis. First, many records do not cover the full time span under study. Thus, the effective number of nodes $N_t^r$ varies with time (see Supplementary Material Fig. S1). Second, while different archives might be significantly affected by same climate variable, they still exhibit both local and proxy-specific effects, so that the outcomes of pair-wise similarity

assessments can be highly case-specific, even though the shared climatic influence might be the same.

In order to address these peculiarities and to simplify the interpretation of the resulting network structures, it appears reasonable to combine spatially close records into *clusters* (as will be further detailed below). In the present context, a *cluster* is a subset of records $C_{t\mathcal{W}}^M \subset \left\{ x_\bullet^i \right\}_{i \in M}$ with $M \subset \{1, 2, \dots N^r\}$ and $\left| C_{t\mathcal{W}}^M \right| \geq 2$. Note that we will consider the assignment of any archive to a specific cluster as being fixed over the entire analysis period (see below). Hence, the existence and size of a given

cluster size vary only due to the (non-) availability of the given archives during different periods of time.

Having obtained a climatologically meaningful grouping of our archives into spatially connected clusters, we define the cross-link density (CLD) between any two clusters $C_{t\mathcal{W}}^K$ and $C_{t\mathcal{W}}^L$ as

$$
\begin{aligned}
CLD_{t\mathcal{W}}^{K,L} \quad &:= \frac{\text{\# links between } C_{t\mathcal{W}}^K \text{ and } C_{t\mathcal{W}}^L}{\text{\# possible links between } C_{t\mathcal{W}}^K \text{ and } C_{t\mathcal{W}}^L} \\
&= \frac{\sum_{i \in K} \sum_{j \in L} A_{t\mathcal{W}}^{i,j}}{\left| K_{t\mathcal{W}} \right| \cdot \left| L_{t\mathcal{W}} \right|}.
\end{aligned} \tag{4}
$$

Note that as we are generally considering evolving (i.e. time-dependent) network structures, the CLD for each pair of clusters will commonly vary in time. However, the CLD values are expected to be more robust tracers of the essential network structure than other commonly used network characteristics, since they combine information from various links and are properly normalized by the (time-dependent) number of records. By making use of this approach, for $S_C$ denoting the number of clusters $\{C_M \dots | | C_M | \geq 2\}$ we can define an $S = \binom{S_C}{2}$ dimensional vector of cross-link densities

$$
\mathbf{X}_{t\mathcal{W}} = \left\{ CLD_{t\mathcal{W}}^{K,L} | K, L \in \{1, \dots S_C\}, K \neq L \right\}.
$$

This way, we effectively obtain coarse-grained networks with fewer nodes (associated with each cluster) and weighted links (CLD values). In turn, we disregard any information on the intra-cluster statistical linkages between individual archives, since



we expect the latter to mainly reflect the intrinsic spatial correlation length of the influencing climate variable. Combining information on intra- and inter-cluster linkages would result in a paleoclimate "network of networks" approach; a framework that has already been employed in a few studies on recent climate variability (Donges et al., 2011; Wiedermann et al., 2016), but might suffer from the low number of nodes in a paleoclimate setting.

## 3.4 Spatial clustering of proxies

As discussed above, it is advisable to simplify the functional paleoclimate network by grouping several archives into spatially connected and climatologically meaningful clusters and study exclusively the temporal changes in the mutual similarity of proxies at the resulting cluster level. Specifically, the obtained clustering should meet the conditions that clusters (i) comprise spatially close archives and (ii) are large enough to reduce the impact of individual records and thus lead to a robust representation of the large-scale spatial co-variability structure.

Given the distinct individual characteristics of the different proxy time series and local as well as archive-specific effects, it is difficult to perform a cluster analysis directly at the set of time series originating from the archives, since such a procedure would likely result in a highly fragmented cluster structure. Instead, given that all proxies in our ensemble are temperature-sensitive, we define clusters as regions which have shown similar inter-annual temperature variability over the modern (instrumental) period. For this purpose, we make use of the gridded ERA-20C reanalysis summer temperature data spanning the whole $20^{th}$ century (Poli et al., 2016). Based upon this data set, we generate a functional climate network reflecting only the strongest absolute linear (Pearson) correlations (as determined by a threshold value $\alpha_C$) among the time series of seasonal mean (annually averaged boreal summer, JJA) temperatures for each grid point over land in the study region. From this network representation, we identify subsets of grid points with high intrinsic and low extrinsic connectivity (referred to as network communities) by applying the so-called Louvain algorithm (Blondel et al., 2008).

We emphasize that the described spatial clustering procedure introduces an additional parameter $\alpha_C$ into the analysis. In general, we observe that small values of $\alpha_C$ yield more but smaller clusters, while larger values lead to a lower number of larger clusters (not shown). One of the main differences between the obtained clusters using different values of $\alpha_C$ (from a reasonable range of values) is the division of Greenland and the border between Central and Eastern Europe. This is also the main difference in using different variables (e.g. ERA-20C winter mean temperature) for the clustering (Supplementary Material Fig. S2).

## 3.5 Statistical modeling by regression

Beyond simple visual analysis of the evolving network structures, we aim to statistically link the obtained time-dependent CLD values with a climate-related variable (in our case, an existing NAO reconstruction by Ortega et al. (2015), in the following referred to as $NAO_{Ortega}$) that reflects a common influence on the co-variability between different regions. The simplest model to establish such a corresponding relationship between $\mathbf{X}_{tw}$ and some variable $Y$ would be a linear model

$$Y = \mathbf{D}_{tw}\mathbf{X}_{tw} + \epsilon_{tw} \tag{5}$$

(c) Author(s) 2017. CC-BY 3.0 License.



with a coefficient vector $\mathbf{D}$ and a noise process $\epsilon_t w$. Note that while in general the CLD values should rather be described as a superposition of different climatic influences, we take here the opposite approach, which is potentially useful for obtaining a reconstruction of the (unknown) climate driver based upon our evolving functional paleoclimate network properties. For a detailed discussion on different (regular vs. inverse) versions of this regression problem and their implications in the context

of paleoclimate reconstructions, we refer to Christiansen (2011) and references therein.

We emphasize that the analysis procedure described above has two free parameters, the threshold values $\alpha_{pr}$ (for generating the paleoclimate network) and $\alpha_C$ (for obtaining the spatial clustering). Given that our general aim is to maximize the inferred information about the mean state of the leading mode of North Atlantic climate variability at inter-annual to multi-decadal scales (probably related to the NAO phase) from the simplified networks, we vary the values of $\alpha_{pr}$ and $\alpha_C$ to obtain an

ensemble of sequences of evolving networks as well as geographical clusterings, each ensemble member corresponding to a different combination of both parameters. For each member, we individually perform a multiple linear ordinary least-square (OLS) regression of all associated CLD values to the 50-year averaged NAO$_{\text{Ortega}}$ reconstruction (Ortega et al., 2015) – in the following denoted as $\overline{\text{NAO}}_{\text{Ortega,50 yr}}$ – using the model in Eq. (5). The parameter combination $(\alpha_{pr}, \alpha_C)$ for which the resulting regression model describes the largest fraction of variability in $\overline{\text{NAO}}_{\text{Ortega,50 yr}}$ (see Supplementary Material Fig. S3) is then

selected for further analysis.

As a final step, we further investigate the linear model (Eq. 5) for $\overline{\text{NAO}}_{\text{Ortega,50 yr}}$ using a Bayesian approach, Markov Chain Monte Carlo (MCMC) regression (Gilks et al., 1995). Unlike OLS regression, this method results not in individual estimates of the different regression coefficients, but in joint distributions for all model parameters. Thereby, we implicitly account for the uncertainty in the description of the target variable. Since some of the considered clusters of paleoclimate archives do not cover

the full Common Era, we can furthermore use the parameter distributions of the full set of clusters as priors to find the new distributions of the reduced set of CLD values, thus utilizing the knowledge of the full data for cases of lower data availability. For performing the MCMC regression, we employ a NUTS sampler (Hoffman and Gelman, 2014) with $10^4$ samples with one quarter of these as burn-in.

## 4   Results

We have followed the procedure described in Sec. 3 to study the evolving paleoclimate networks derived from the set of 37 paleoclimate records described in Sec. 2. The window size $\mathcal{W}$ of our analysis has been selected as 50 years, consistent with the averaging window in $\overline{\text{NAO}}_{\text{Ortega,50 yr}}$. We have considered sliding windows with a mutual offset of 1 year, implying an overlap of 49 years between subsequent windows. The threshold values $\alpha_{pr}$ and $\alpha_C$ have been determined by maximizing the explained variance of the 50-year averaged NAO$_{\text{Ortega}}$ reconstruction (see Supplementary Material Fig. S3), yielding $\alpha_{pr} = 0.46$ and

$\alpha_C = 0.0104$. The resulting spatial clusters of archives used in the analysis are displayed in Fig. 3.

Figure 4 shows the simplified networks and dominating cross-cluster links for some exemplary time windows, using a lower threshold value of $\alpha_{pr} = 0.1$ to better highlight the strongest correlations (for illustrative purposes only). The most informative





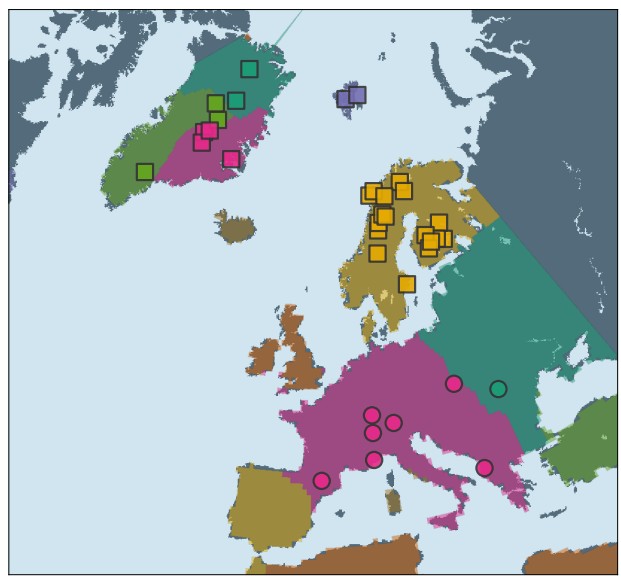

**Figure 3.** Division of the study area as obtained by cluster analysis of the ERA-20C summer mean temperatures, together with the paleo-climate archives used in this study ($\alpha_C \approx 0.01$). Disconnected regions of the same color correspond to different clusters (cf. the different markers for the paleoclimate archives).

clusters are located in Southern Greenland (SG), Fennoscandia (FS) and Central Europe (CEU) and cover all of the Common Era.

During the first millennium CE, we can distinguish two common, qualitatively different states of the network, one being dominated by connections between FS and the other two clusters (Fig. 4a,c) and another exhibiting strong correlations between archives from SG and CEU (Fig. 4b,d). During the second millennium CE (with considerably more archives available), we more clearly identify periods during which mainly West-East connections between Greenland (G), Svalbard (S) and FS are present (Fig. 4f,i). During other times, North-South connections involving CEU are more strongly expressed (Fig. 4 e,g,h). The latter is commonly the case during time intervals for which $\overline{\text{NAO}}_{\text{Ortega,50 yr}}$ indicates a negative mean reconstructed NAO index, while cross-cluster links are more concentrated within the northern North Atlantic sector during positive NAO phases.

This general finding is further supported by the mean regression coefficients of the linear model (Eq. 5) for $\overline{\text{NAO}}_{\text{Ortega,50 yr}}$. The respective strengths and signs of the most relevant regression coefficients are illustrated in Fig. 5 and summarized in Supplementary Material Tab. S2. The time-evolution of the six CLDs associated with the largest coefficients are shown in Supplementary Material Fig. S4. The corresponding results further demonstrate that the presence of strong West-East connections is indicative of a positive NAO phase, while North-South connections, especially between CEU and the rest of the network, point toward negative NAO phases. The largest regression coefficients correspond to CLDs between SG and FS and CEU.







(a) 150 to 200 CE

(b) 350 to 400 CE

(c) 650 to 700 CE

(d) 710 to 760 CE

(e) 1100 to 1150 CE

(f) 1200 to 1250 CE

(g) 1400 to 1450 CE

(h) 1620 to 1670 CE

(i) 1896 to 1946 CE

**Figure 4.** Simplified functional paleoclimate networks for different exemplary time windows.





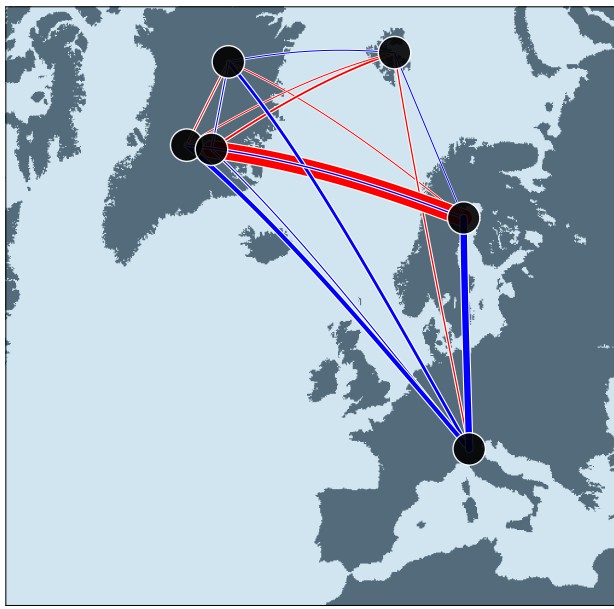

**Figure 5.** Regression coefficients relating the 15 cross-link densities to $\overline{\text{NAO}}_{\text{Ortega,50 yr}}$. Red (blue) colors indicate positive (negative) sign of the coefficient, whereas the width of the drawn links is proportional to the mean coefficient value.

As an additional test for the validity of our estimated linear model describing multi-decadal NAO variability, we split the $\overline{\text{NAO}}_{\text{Ortega,50 yr}}$ reconstruction into two parts of equal size, using one part as training period and the other for validation. We apply OLS regression of the cross-link densities to $\overline{\text{NAO}}_{\text{Ortega,50 yr}}$ during the training period and then compare the values predicted by the obtained model for the validation period with the actual values of $\overline{\text{NAO}}_{\text{Ortega,50 yr}}$. Using both parts as respective training and validation periods, the resulting $r^2$ values are very low ($0.15$ and $0.28$, respectively). Hence, the linear model can scarcely explain the amplitude of the supposed long-term average NAO variability as expressed by $\overline{\text{NAO}}_{\text{Ortega,50 yr}}$. Nevertheless, the obtained sign of the NAO phase is identified correctly in $68\%$ and $71\%$ of the considered time windows covered by $\overline{\text{NAO}}_{\text{Ortega,50 yr}}$, respectively, indicating that our results still have a certain value, as will be further detailed below. In the following, we will refer to this quantity as the true sign ratio (TSR). Notably, taking the second (more recent) half of $\overline{\text{NAO}}_{\text{Ortega,50 yr}}$ as regression period (which corresponds to a period with more records than the first one) results in higher values of both, $r^2$ and TSR. This finding suggests, that using additional records, which do not have data outside of the regression period, can still lead to a better performance of our model due to a better interpretation of the existing links. Future work along the lines of the present paper might explicitly utilize this observation in a Bayesian analysis framework.

In order to better understand the remaining $\sim 30\%$ of time intervals during which the sign of $\overline{\text{NAO}}_{\text{Ortega,50 yr}}$ is not correctly represented by our model, we perform an additional type of cross-validation, this time independently leaving out consecutive 50-year windows as validation periods and taking the rest of the data for model estimation. The results of this analysis are shown in Supplementary Material Fig. S5. We find, that whenever the TSR in a given validation periods is clearly lower than



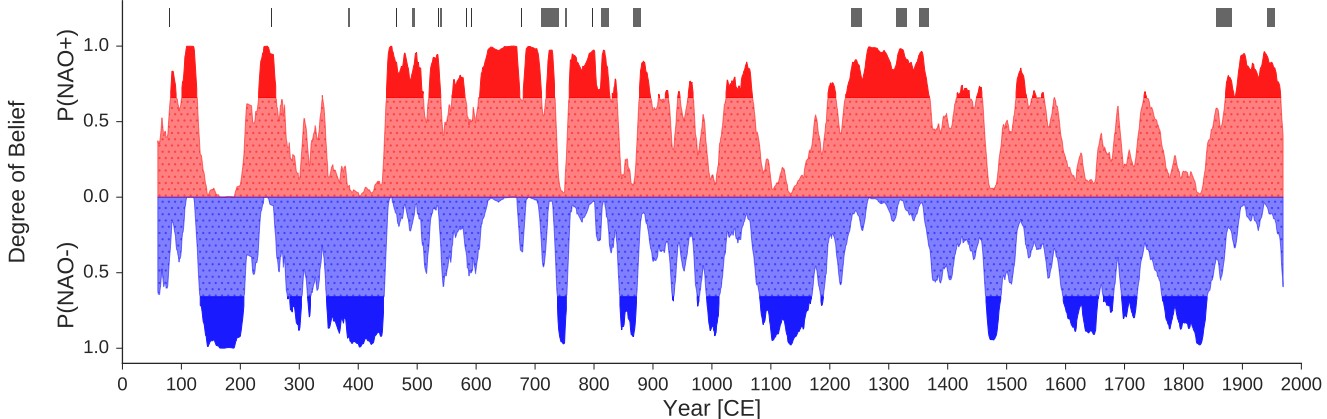

**Figure 6.** Degree of belief that the NAO is in a specific (positive vs. negative) phase during each 50-year time window. The figure has been smoothed by a 10-year moving-average filter to enhance its readability. The brighter areas indicate time intervals for which less than 66% of all considered MCMC ensemble members agree upon the sign of the reconstructed NAO variable. Gray bars correspond to known major drought episodes in the Western Mediterranean as discussed in Sec. 5.

the mean value of 0.69, $\overline{\text{NAO}}_{\text{Ortega,50 yr}}$ is either consistently close to zero or exhibits a transition between the two phases. In turn, during periods of strong, persistent positive or negative NAO phases in $\overline{\text{NAO}}_{\text{Ortega,50 yr}}$, the TSR exhibits very high values. Nevertheless, there are some periods during which our model differs markedly from $\overline{\text{NAO}}_{\text{Ortega,50 yr}}$, e.g. in the 17th century. Hence, we conclude that if our model is used for the purpose of hindcasting the (themselves statistically reconstructed)

NAO values according to Ortega et al. (2015), this might result in incorrect identifications of the mean NAO phase at values where $\overline{\text{NAO}}_{\text{Ortega,50 yr}}$ is close to zero, since under these conditions, our ensemble of equally likely NAO "trajectories" obtained from MCMC regression includes both, positive and negative estimates for the corresponding time window. In turn, the model performs well in correctly identifying strong and persistent positive and negative NAO phases. However, we observe that the actual timing of transitions between distinct NAO phases can differ between $\overline{\text{NAO}}_{\text{Ortega,50 yr}}$ and our model (Supplementary

Material Fig. S5).

Despite the fact that only four geographical clusters of paleoclimate archives cover the full Common Era, our model allows to qualitatively expand the existing "smoothed" NAO reconstruction by Ortega et al. (2015) over the last two millenia and thus obtain relevant information about the dominant NAO phase at multi-decadal time scales. To do so, we draw 10,000 realizations of the regression coefficient distributions of our model and calculate the corresponding NAO values for each point

in time based on the available CLDs. From these realizations, we determine a degree of belief, that the NAO was in a specific dominant phase at a given time window. The results of this probabilistic description are shown in Fig. 6. We find that during the Common Era, there have been several phases during which the multi-decadal NAO variability was preferentially in a positive phase (e.g. during the migration period and the late medieval times), which alternated with strong negative phases (e.g. during



the Little Ice Age, LIA) or intervals with generally more variability (e.g. the late Roman period or the centuries around 1000 CE) indicating more unstable conditions of the large-scale atmospheric circulation over the North Atlantic region.

Finally, we have additionally tested the robustness of the estimated regression model by varying $\alpha_C$ over a reasonable range (similar variations of $\alpha_{pr}$ were found not to alter the obtained results markedly, which is not explicitly shown here).

Supplementary Material Fig. S6 shows the corresponding results in terms of OLS-based regression models obtained with different parameter values. While most parameter sets close to the selected optimal one yield very similar results, there are few exceptions demonstrating the importance of this sensitivity analysis. A particularly remarkable example is found in the second half of the 5[th] century, where a transition from a predominantly negative NAO phase to a positive one is observed, the exact timing of which, however, differs significantly among the different regression models. This observation underlines, that

our model has some uncertainty – not only in terms of reconstructed values, but also the timing of changes – which cannot be accounted for outside an ensemble-based approach.

## 5   Discussion

### 5.1   Climatological interpretation

In our analysis, we have related evolving functional paleoclimate networks to the dominant mode of multi-decadal variability

of the atmospheric circulation in the North Atlantic region as reported by Ortega et al. (2015), which has been associated with long-term changes of the NAO. As seen from Figs. 4 and 5, the number of statistically relevant connections between Greenland, Svalbard and Fennoscandia is enhanced during positive phases of the NAO (according to the reconstruction by Ortega et al. (2015)), whereas links involving Central Europe are more pronounced during negative NAO phases. Thus, we interpret strong West-East correlations in the study region as indicators of a positive NAO phase, while North-South connections point to

negative NAO phases. This observation can be qualitatively addressed by visualizing the obtained networks as in Figs. 4 and 5. More detailed statements can be made based upon a systematic inter-comparison between the time-dependent CLDs associated with distinct cluster pairs, as well as the evaluation of the regression to the existing reconstruction $\overline{\mathrm{NAO}}_{\mathrm{Ortega,50\ yr}}$ in terms of the linear model (Eq. 5). The latter analysis helps relating different NAO phases to certain cross-cluster links in a more objective fashion.

The interpretation of a preferred presence or absence of certain CLDs during specific NAO phases agrees well with the known NAO impact on European climate variability during the instrumental period. A positive NAO phase is commonly related to a northward shift of the westerlies, which causes milder temperatures and stronger precipitation in Northern Europe during boreal winter. Thus, from the observation that tree ring chronologies are strongly influenced by intense winter precipitation, it is reasonable that the considered archives from Fennoscandia (Central Europe) are particularly affected by positive (negative)

NAO phases. The non-stationary influence of winter conditions is further illustrated in Supplementary Material Fig. S7. While many of the ice core records from Greenland, which were instrumental in obtaining the $\mathrm{NAO}_{\mathrm{Ortega}}$ reconstruction, exhibit strong negative correlations with that reconstruction throughout the last millennium, there is much more variability in the correlations with $\mathrm{NAO}_{\mathrm{Ortega}}$ for all the other records.





Suppose that we are given a reference time series which exhibits a stationary relationship with the "true" NAO (in our case, the aforementioned Greenland ice cores). If the variability of any particular record shows a strong similarity with this reference series during a specific time, we expect that this record carries significant information about the NAO phase. As the actual imprint of the NAO is different for different regions, this information is present in different proxy groups at different times. In

our case, the Greenland ice cores act as a filter to indicate which regions are strongly influenced by the NAO and, thus, which NAO phase is more probable.

The final MCMC regression model correlates well with $\overline{\mathrm{NAO}}_{\mathrm{Ortega,50\,yr}}$ ($r^2 = 0.58$). However, this could simply result from overfitting, as indicated by our cross-validation. While the obtained quantitative values of the reconstructed NAO index of Ortega et al. (2015) are thus not reliably described by our model, the latter performs well in resolving the dominant NAO phase.

Drawing upon the knowledge about relationships between certain cross-cluster links and the NAO phase as discussed above, it is generally possible to extend the existing (smoothed) NAO index reconstruction $\overline{\mathrm{NAO}}_{\mathrm{Ortega,50\,yr}}$ to the entire first millennium. However, since our linear model is not capable of describing the amplitude variability of $\overline{\mathrm{NAO}}_{\mathrm{Ortega,50\,yr}}$ adequately, this should only be considered as qualitative information about the likely NAO phase.

While there is an influence of the NAO on regional temperatures at multi-decadal time scales, strong low-frequency tem-

perature variations are largely associated with solar activity changes and explosive volcanism (Crowley, 2000). Thus, the observation of elevated temperatures reported for most of the Roman Warm Period (RWP) and Medieval Climate Anomaly (MCA) as opposed to lower temperatures during the Late Antique (Luterbacher et al., 2016; PAGES 2k Consortium, 2013) does not contradict our qualitative reconstruction of the predominant NAO phase. Instead, there might be common causes for such apparently contradictory observations. For example, explosive volcanism has been discussed as a major driver of the Late

Antique Little Ice Age climate (Büntgen et al., 2016), but is also known to frequently trigger positive NAO-like atmospheric dynamics during the years following strong eruptions (Robock, 2000). Even though this is mostly a short-term effect, a high frequency of strong eruptions, as present during the Late Antique (Sigl et al., 2015), might have had a more persistent influence.

We emphasize that there are certain limitations to the usage of CLDs and our linear model to draw conclusions about the predominant NAO phase. First, most of the CLDs exhibit downward trends and progressively decreasing variance throughout

the Common Era (Supplementary Material Fig. S4), which is probably related to the lower number of records as one goes back in time. This might add a considerable bias to any application of our methodological framework extending further back in time than the last millennium. In our case, this effect might favor positive NAO phases, since the regression coefficient for the connection between Southeast Greenland and Fennoscandia is by far the largest among all coefficient. Furthermore, our regression is based on a proxy-based reconstruction, which contains large uncertainties itself and therefore has limited

value as a "ground truth". Ortega et al. (2015) reported, that their reconstruction explains only about $40\%$ of the variance of the observed NAO. In turn, the explanatory value of the time series ($\overline{\mathrm{NAO}}_{\mathrm{Ortega,50\,yr}}$) upon which our regression analysis is based, is a key assumption beyond our procedure. Moreover, our cross-validation showed that the linear model can disagree with $\overline{\mathrm{NAO}}_{\mathrm{Ortega,50\,yr}}$ especially in cases in which that reconstruction has values close to zero, as well as in the timings of some transitions between positive and negative NAO phases. This intrinsic uncertainty of our qualitative reconstruction has been



addressed by using MCMC regression to take a probabilistic view on the NAO phase. At time periods where the reconstructed $\overline{\text{NAO}}_{\text{Ortega,50 yr}}$ is close to zero, no particular phase is preferred in general (Fig. 6).

Following upon the considerable uncertainties in using our linear model to obtain qualitative estimates of the NAO phase during the entire Common Era, we will next compare our corresponding results to other long-term NAO-related climate recon-
structions. Moreover, we will utilize further independent information in terms of documented drought periods for an independent validation of our reconstruction. Finally, we will discuss how long-term NAO variability might have affected European societies during the first millennium CE.

## 5.2   Comparison with other NAO reconstructions

Our model is able to reproduce most features of the $\overline{\text{NAO}}_{\text{Ortega,50 yr}}$ reconstruction, including a dominant positive NAO phase
during the late MCA, generally stronger variability during the LIA, with a tendency towards a more negative NAO phase, and another strongly positive phase during the 20th century, which is also in accordance with instrumental records (Vinther et al., 2003).

In contrast to other previous findings by Trouet et al. (2009), we do not observe strong West-East correlations during the early MCA, which suggests that this time interval has not been characterized by a strongly positive NAO.
Another recent NAO reconstruction by Olsen et al. (2012) used a more than 5000 years-long lake record from Southern Greenland to trace the predominant NAO phase during the entire Late Holocene. Our mean model correlates only extremely weakly with their reconstruction during the common period ($r^2 = 0.04$). One probable reason for this disagreement could be time uncertainty in their very long-term reconstruction, which Olsen et al. (2012) report as being of multi-decadal order during the first millennium CE. Furthermore, their record has been adjusted to the reconstruction by Trouet et al. (2009), which
disagrees with NAO$_{\text{Ortega}}$ at many times.

Finally, it is worth considering a recent study by Deininger et al. (2016) who used 11 European speleothem records and analyzed their mutually coherent dynamics, which can be connected with changes in the North Atlantic circulation regimes reflecting long-term NAO variability. Their results indicate a strong, persistent positive NAO phase during the entire MCA and a tendency towards a negative NAO phase during the LIA, the first being in partial disagreement and the second in accordance
with the NAO$_{\text{Ortega}}$ reconstruction. In addition, Deininger et al. (2016) report a dominant negative NAO phase between about 250 and 500 CE and a neutral-to-positive NAO phase thereafter. This observation agrees with our qualitative extension of $\overline{\text{NAO}}_{\text{Ortega,50 yr}}$.

## 5.3   Comparison with historical droughts

Winter NAO and the corresponding precipitation anomalies exhibit known linkages to droughts in many parts of Europe,
most significantly in the Western Mediterranean region (López-Moreno and Vicente-Serrano, 2008; Cook et al., 2016). Due to their severe impacts on agricultural productivity, droughts are some of the best documented weather extremes across historical times. Thus, existing reports of historical drought periods can be used as an independent source of information to test the consistency of our qualitative NAO reconstruction. For this purpose, we use three accounts of droughts during the Common Era.



McCormick et al. (2012a) collected climatic evidence from the period of the Roman Empire (up to 800 CE) and reported 8 large droughts in the Western Empire (accessible through McCormick et al., 2012b). Domínguez-Castro et al. (2014) summarized historical evidence from Muslim sources for Southern Spain, a region exceptionally vulnerable to NAO-related droughts (Cook et al., 2016), from 711 to 1010 CE. They identified three major drought periods during this time. In addition, Cook et al. (2016)

discussed droughts during the last millennium in the Mediterranean region based upon the Old World Drought Atlas (OWDA) (Cook et al., 2015). They reported that the drought index constructed for the Western Mediterranean correlates well with NAO$_{Ortega}$. Therefore, we consider here only the five strongest drought events as discussed in Fig. 5 of their paper.

It has to be noted that although droughts are strongly related to precipitation deficits potentially associated with the dominant NAO phase, they are complex phenomena with multiple causing factors. Thus, we do not expect the timing of all droughts in

the considered region to be fully explained by any particular NAO reconstruction.

In Fig. 6, the major drought periods discussed in the aforementioned publications are marked as gray bars. Comparing the timing of these periods with our qualitative reconstruction of the NAO phase, we find that most droughts indeed coincide with positive NAO phases. However, a larger number of reported droughts during a specific time period does not necessary imply a higher frequency of droughts. In case of historical documents, an increase in reported events could also indicate that a society

was more vulnerable to the impacts of droughts and, thus, found them more worth reporting.

### 5.4 Possible impacts on human societies

As mentioned in the previous section, it can be expected that at longer time scales, the alternation between different phases of the NAO has had a considerable impact on human societies via modifications of temperature and precipitation patterns and their resulting consequences for natural and agricultural ecosystems (Hurrell et al., 2003; Hurrell and Deser, 2010, and

references therein). In the following, we discuss possible implications of our qualitative reconstruction of the NAO phase in the context of European history during the Common Era. Since the climatic influence of the NAO differs among different parts of Europe, we restrict this discussion to two key regions, the Western Roman Empire and Norse colonies in the North Atlantic. Prior to presenting some further thoughts on corresponding relationships, we emphasize that one has to keep in mind, that climatic conditions have almost never been the sole reason for societal changes. However, they can be either beneficial or

disadvantageous, also depending on how vulnerable a society is to environmental disruptions (Diaz and Trouet, 2014; Weiss and Bradley, 2001; Diamond, 2005). Because of the complex interrelationship between human societies and environmental and climatic factors (Engler, 2012; Engler and Werner, 2015), discussions about any possible causal links would in general be highly speculative, so that we explicitly refrain from making any corresponding claims.

Our qualitative NAO reconstruction describes a prevalent negative NAO phase with shorter interruptions by positive phases

until about 450 CE. In the Western Mediterranean, such a climatic setting commonly corresponds to milder and wetter winters (Hurrell et al., 2003). This expectation is in line with previous descriptions of this period as warm (Luterbacher et al., 2016) and humid (García et al., 2007; Martín-Chivelet et al., 2011; Desprat et al., 2003), even though there is considerable disagreement about the exact timing of the termination of this phase among different paleoclimate archives. The aforementioned conditions might have generally been beneficial for the Western Roman Empire (McCormick et al., 2012a). In turn,



decreasing temperatures, together with more frequent droughts following the RWP might have progressively added stress to societies in the Western Mediterranean, which had already been weakened by internal conflicts, plagues, invasions and other factors at this time (McCormick et al., 2012a; Diaz and Trouet, 2014). López-Moreno and Vicente-Serrano (2008) describe that a corresponding effect could have played a key role not only in the Western Mediterranean, but also in the Northern Balkan

region. Thus, during the prolonged negative phase in the 4[th] and 5[th] century CE, these regions received higher precipitation and might thus have been a target for invading Huns, who were possibly aiming to escape drought conditions in Central Asia (McCormick et al., 2012a) triggering the mass migration of the Late Antique (Halsall, 2007).

Patterson et al. (2010) discussed the impact of North Atlantic seasonality on Norse colonies based upon $\delta^{18}$O values of near-shore mollusks from Iceland, which trace changes in the ocean circulation potentially accompanied by certain preferred NAO

patterns. Specifically, they reported cold periods around 410 CE and between 1380 and 1420 CE, while warm temperatures are noted from 230 BCE to 140 CE and around 600 to 1000 CE. The latter is consistent with findings by Werner et al. (2017), who date the maximum of the MCA in the Arctic to the period between about 960 and 1060 CE, which is in line with our qualitative NAO reconstruction. A positive NAO phase during the second half of the first millennium would have lead to generally warmer temperatures and less sea ice and would thus have been favorable to marine ecosystems in the region (Hurrell et al., 2003).

While a positive NAO phase was most likely beneficial for settlement on, and sustained population of Iceland, it was also associated with enhanced storm activity and increased wave heights in the North Atlantic (Serreze et al., 1997; Bader et al., 2011), rendering the sailing conditions more difficult. The first evidence of settlements on Iceland dates back to around 870 CE, a time at which our reconstruction indicates a short period of negative NAO phase. This implies, that while the overall positive NAO phase was helpful for establishing these settlements, short negative NAO phases were supportive for longer

expeditions across the North Atlantic. Settlement on Greenland follows a similar pattern, being established during the late 9[th] century, which again corresponds to a period of probably negative NAO phase. The situation differs from those in Iceland in that negative NAO phases would be accompanied by warmer temperatures in Greenland. Thus, the negative NAO phases during the MCA could have been beneficial for settlement in Southern Greenland. In turn, during the 14[th] century – a phase of prolonged positive NAO and, thus, lower temperatures – most of the Norse colonies on Greenland were abandoned, possibly

affected by the associated environmental changes (Diamond, 2005).

In summary, some key periods of the history of those parts of Europe that are most strongly exposed to long-term variations of North Atlantic climate appear closely related with environmental changes that are consistent with our qualitative NAO reconstruction, providing additional evidence for the general validity of the obtained long-term patterns.

## 6   Conclusions

In this study, we have demonstrated that functional networks based on paleoclimate proxy records from multiple, spatially distributed archives offer great potentials for identifying spatial patterns of atmospheric circulation in the European North Atlantic sector together with information on their associated long-term variability.



Specifically, we have obtained a new 2000 year-long qualitative reconstruction of the leading mode of regional inter-annual temperature variability most likely associated with multi-decadal NAO variability. By combining visual inspection of changing patterns in the (coarse-grained) network representations with a simple linear regression model, we have presented a climatologically consistent interpretation of the time-dependent strength of correlations between groups of proxies from different

parts of Europe and the northern North Atlantic as indicators of different NAO phases. In general, we relate strong East-West connections with a positive NAO phase and North-South connections with a negative phase. While the linear model does not trace the exact variability of the reconstructed NAO index by Ortega et al. (2015) very well, it still provides a good qualitative explanation of the succession of different phases at multi-decadal time scales. Uncertainties arise mainly from an insufficient description of the observations by the linear model, a possible bias induced by a decreasing number of records when going fur-

ther back in time, and existing uncertainties in the $\overline{\text{NAO}}_{\text{Ortega,50 yr}}$ time series, upon which the regression is based. Thus, future consideration of additional high-resolution paleoclimate records from the North Atlantic region, especially from regions like Svalbard, Greenland and Eastern Europe, might further improve the model fit substantially. Notably, using standard similarity measures like the Pearson correlation coefficient, it is not feasible to use shorter time windows than the 50-year windows used in the present analysis. Thus, our approach cannot yet be directly applied to the instrumental record as regression target. The

use of different similarity measures for the purpose of the present study should be further addressed in future work.

Our qualitative expansion of the $\overline{\text{NAO}}_{\text{Ortega,50 yr}}$ reconstruction demonstrates, that the Common Era has been characterized by time periods with different behavior of the NAO. In general, multi-decadal changes of the predominant NAO phase occurred relatively frequent during Roman and early medieval times, while there have been other periods characterized by a persistent positive or negative NAO phase. The first is the case for most of the migration period, the late medieval times and the 20[th]

century, while the latter is found during the late Roman times and the Little Ice Age. These long-term changes in the NAO phase might have had a considerable impact on European societies, as the NAO phase is associated with the likelihood of regional droughts as well as precipitation and temperature extremes, which may have directly affected agricultural productivity. In this spirit, specific phases might have supported some European societies while negatively affecting others.

**Code availability**

All calculations in this work have been based upon open source software. AAFT surrogates have been generated using the Python package `pyunicorn` (Donges et al., 2015). Cluster analysis of reanalysis data has been conducted using the `community` package. MCMC regression has been performed with the `pyMC3` package (Salvatier et al., 2016).

**Data availability**

Most of the time series used in this study are available through the PAGES 2k project. The obtained qualitative NAO recon-
struction is provided as part of the Supplementary Material accompanying this paper.



*Author contributions.* JGF designed and conducted the analysis and prepared the manuscript. RVD designed and supervised the analysis. JPW advised the data selection. RVD and JPW critically revised the manuscript and the interpretation of the obtained results.

*Competing interests.* The authors declare no conflict of interest.

*Acknowledgements.* This work has been financially supported by the German Federal Ministry for Education and Research (BMBF) via the

5    BMBF Young Investigators Group "CoSy-CC$^2$ - Complex Systems Approaches to Understanding Causes and Consequences of Past, Present and Future Climate Change" (grant no. 01LN1306A), and by the joint German-Norwegian project "Nonlinear variability and regime shifts in Late Holocene climate: regional patterns and inter-regional linkages in multi-proxy networks and climate simulations" jointly funded by the German Academic Exchange Service (DAAD project no. 57245873) and the Research Council of Norway. RVD acknowledges additional support by a Bjerknes Visiting Fellow grant. The authors thank Dmitry Divine for fruitful discussions stimulating the developments described

10   in this work. The data used in this study have been kindly provided by the Arctic2k group of the PAGES2k initiative and Saija Saarni.





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
