# Peer review of "Reconstructing Late Holocene North Atlantic atmospheric circulation changes using functional paleoclimate networks"

_Climate of the Past, 2017_

## Referee Comment (RC1) · Anonymous Referee #1 · 7 Apr 2017

The authors use a network approach to analyze the connections in a set of paleo-records in the North Atlantic region. The connectivity in the network is then related to a previous NAO reconstruction. This relation is used to expand the NAO reconstruction back in time.

The improvement of reconstruction techniques is an important subject and the NAO is a dominant mode of variability. I don't doubt that the authors have a deep knowledge of networks and that the analysis is well performed. However, there are many steps in the analysis which are not familiar to the average reader. I will strongly suggest that the authors try – wherever possible – to relate the network properties to more physical properties. If the paper gets too long they could delete section 5.4 which seems a bit

out of topic.

So, as I see it. the paper certainly deserves to be published but it could benefit from a more pedagogical approach.

Major comments:

1) While I in general find that the paper is well written I also find that it is very technical. The analysis includes several steps and it is not always easy to see the physical content. For example, what does Fig. 5 actually mean? It looks a little as the impact of the NAO on the temperature; negative correlations between the NAO and temperatures in middle Europe and positive correlations in Greenland and Scandinavia. But I guess it is more a picture of how tele-connection (or coherent?) patterns depend on the NAO which in itself can be seen as a tele-connection? Perhaps the authors could use observed temperatures to demonstrate how the tele-connection patterns look in the two phases of the NAO? The spatial coherence is already used for the spatial clustering of proxies.

More generally, I think it would be good if the authors tried (even more) to relate the network results to quantities of a more simple and well-known character.

2) I noticed that the reconstruction does not perform well regarding the correlation in the cross-validation test. Nonetheless, the authors use it to predict the sign of the NAO for which the method seems to be correct about 70 % of the time. I don't really understand the explanation the authors give (p13). It would help if a figure of the Ortega reconstruction and the new reconstruction was shown.

As for the comparison of the present reconstruction with other reconstructions it should be noted that both the reconstruction methodology and the proxy selection will be important. I would suggest that the authors produce a reconstruction from their proxies using a simple multiple regression scheme between the NAO and the proxies. This might help getting an idea of which improvements the network method actually brings.

3) Section 3.2: It seems that the similarity is defined from the p-vales alone. In my understanding it should be based on a combination of the size of the correlation and the p-value. As a large correlation can be insignificant so can a small p-value be connected to a weak correlation.

The similarity does not seem to take the sign of the correlation into account. From a physical point of view there is a big difference if two point are positively or negatively correlated. So is not a lot of information lost in this process?

4) Introduction, page 2: Networks probably have some advantages in some situations. However, networks were developed for studies of discrete phenomena such as those in sociology. In the study of climate we deal with fields that are continuous in both space and time. It therefore seems backwards to reduce the problem to a network. We must loose information that other methods based on fields take into account. I know that the present paper is not the place for a philosophical discussion but the concern could be mentioned.

Minor comments:

Is the A in Eq. 1 used anywhere?

Caption to Fig. 4 should be improved.

Page 7, top: I don't see how the AAFT procedure can be applied to the 4 incomplete proxies. The AAFT includes a Fourier transform.

Fig. 5: The bright areas are not easy to see. By the way: Is CE an accepted standard? It always takes me a while to figure out the direction of the axis.

P9,l5: More recent and complete references are Christiansen 2014 (10.1175/JCLI-D-13-00299.1) and Christiansen and Ljungqvist 2017 (10.1002/2016RG000521).

---

## Referee Comment (RC3) · Anonymous Referee #3 · 18 Apr 2017

In this paper, the authors use a network approach to investigate climate teleconnections across the North Atlantic region during the Common Era and relate climate in this region to the NAO. The authors take an interesting approach toward utilizing published paleoclimate records for reconstructing a regionally important climate index. I agree with the other Referees that this work should be published with some modifications outlined below.

I am in agreement with Referee 1 who suggests that the authors take a more "pedagogical" approach toward describing their methodology. Whenever possible, relating the purpose of equations in words as well, providing definitions for all variables, and including a table of variables that readers can refer back to would help to clarify the

approach taken by the authors. This will make their work more accessible and thus their approach will more likely be followed by others in the future.

I suggest that the authors include more discussion of the proxies they are including in their analysis, which archives they come from, and what the records are interpreted to show across the interval in question. If the authors are not space limited, I would suggest moving the table of proxies into the main text so that readers can clearly see what records are being used and so the original citations for the records can be included in the main text citations. As this is only 37 records, it does not seem unreasonable to include in the main text. I also suggest a more careful description of the clustering of sites with some information to validate this approach – to show that each proxy does reflect regional temperature to a reasonable degree and can be clustered with other sites in the region. Some discussion of the uncertainties involved in the proxy records, age uncertainty as well as uncertainty with what each proxy reflects, should also be included. I agree with Referee 1 that to generate space to accommodate clarifications, the section 5.4 could be reduced or removed as it seems overly speculative.

I also suggest including a plot of the Ortega NAO reconstruction as well as a description of how this was constructed and what records went into it and any potential overlap with the records used in this analysis. Something along the lines of Figure S5 would be useful in the main text to show the comparison between the reconstructions generated here and the Ortega reconstruction.

Minor comments:

Need to explain symbology for site markers – changes from figure to figure, not sure what it means. This is true for both the main text and supplementary figures (e.g. Figure S2)

Figure 4 caption needs more description – what do the line thicknesses represent? Why were these time intervals chosen? What controls when points are shown or not shown?

Elaborate on what is meant by most "informative" clusters and why this is the case (page 9, line 32 – page 10, line 2).

Are the Deininger et al., (2016) records (mentioned line 25 of page 16) included in the analysis? If not, why not? A diagram of the reconstruction presented here, the Deininger work, and the Ortega reconstruction may be informative.
* * *

---

## Short Comment (SC1) · 5 May 2017

The PAGES Data Stewardship Integrative Activity seeks to advance best practices for sharing data generated and assembled as part of all PAGES-related activities. As part of this activity, a team of reviewers has been constituted for the "Climate of the Past 2000 years" Special Issue. The data team is reviewing the data handling within each of the CP-Discussion papers in relation to the CP data policy and current best practices. The team has identified essential and recommended additions for each paper, with the goal of achieving a high and consistent level of data stewardship across the 2k Special Issue. We recognize that an additional effort will likely be required to meet the high level of data stewardship envisaged, and we appreciate dedication and contribution of the

authors. This includes the use of Data Citations (see example in supplement). We ask authors to respond to our comments as part of the regular open interactive discussion. If you have any questions about PAGES Data Stewardship principles, please contact any of us directly.

Best wishes for the success of your paper,

2k Special Issue Data Review Team (Darrell Kaufman, Nerilie Abram, Belen Martrat, Raphael Neukom, Scott St. George) and ex-officio team members (Marie-France Loutre, Lucien von Gunten)

Essential additions for this paper:

(1) Expand the "data availability" section to include details on where the input and output data are archived. A Data Citation is needed for the output data (#6 below).

(2) Add Data Citations for all of the proxy datasets listed in Table S1.

(3) For those data not already in a public repository, submit essential metadata along with the time series and include the Data Citation in Table S1.

(4) For those records with previous PAGES 2k IDs, include cross references to those IDs in Table S1 (see Table 1 in PAGES 2k Consortium (in press) for PAGES2k IDs). Also, the PAGES2k temperature database version number that was used must be stated in "data availability" and in the title to Table S1.

(5) Explain why records from the PAGES 2k database that meet the criteria for this study appear to have been excluded from this analysis.

(6) Submit the primary outcome of the data analyses to a public repository. This includes: (a) The NAO index reconstruction (Fig S6). (b) The cross-link density time series in Fig. S4.

Recommended elements are:

[Figure]

(7) Archive the values for the NAO reconstruction in 50-year bins that was used to correlate with the output of this study (last row in Fig. S7; from Ortega et al., 2015)

Please also note the supplement to this comment:
http://www.clim-past-discuss.net/cp-2017-41/cp-2017-41-SC1-supplement.pdf

---

## Author Comment (AC1) · 15 Jun 2017

*The authors use a network approach to analyze the connections in a set of paleo-records in the North Atlantic region. The connectivity in the network is then related to a previous NAO reconstruction. This relation is used to expand the NAO reconstruction back in time. The improvement of reconstruction techniques is an important subject and the NAO is a dominant mode of variability. I don't doubt that the authors have a deep knowledge of networks and that the analysis is well performed. However, there are many steps in the analysis which are not familiar to the average reader. I will strongly suggest that the authors try – wherever possible – to relate the network prop-*

*erties to more physical properties. If the paper gets too long they could delete section 5.4 which seems a bit out of topic.*

In many cases, giving direct physical (climatological) interpretations of specific network characteristics is difficult. For example, a link between two records (nodes) does not necessarily imply similar "internal" temperature variability, but could also originate from a similar influence of extremes in winter precipitation (for some proxies) or a shared external forcing. Links between clusters are just aggregated connections between climatologically meaningful regions with the aim of defining and calculating a more robust measure that minimizes local effects. However, it is still difficult to directly attribute a strong linkage to a physical property. We will try to explain the meaning of network properties in each step in more detail in our revised manuscript and hope that this will make it clearer to a general audience.

*So, as I see it. the paper certainly deserves to be published but it could benefit from a more pedagogical approach.*

We thank the reviewer for this positive overall recommendation.

*1) While I in general find that the paper is well written I also find that it is very technical. The analysis includes several steps and it is not always easy to see the physical content. For example, what does Fig. 5 actually mean? It looks a little as the impact of the NAO on the temperature; negative correlations between the NAO and temperatures in middle Europe and positive correlations in Greenland and Scandinavia. But I guess it is more a picture of how tele-connection (or coherent?) patterns depend on the NAO which in itself can be seen as a tele-connection? Perhaps the authors could use observed temperatures to demonstrate how the tele-connection patterns look in the two phases of the NAO? The spatial coherence is already used for the spatial clustering of proxies.*

Figure 5 shows the correlation of the connectivity between groups of records with the NAO reconstruction by Ortega et al. This is not directly a (temperature related) teleconnection between different regions, but potentially influenced by secondary effects, like winter precipitation extremes in tree ring records, as well. More explicitly, thick red lines indicate that positive NAO phases coincide with relatively many cross-cluster links, the number of which is reduced during negative NAO phases. For blue links, the relation is just the opposite (more links during negative NAO phases than during positive ones).

*More generally, I think it would be good if the authors tried (even more) to relate the network results to quantities of a more simple and well-known character.*

Please, see the response to the general comment.

*2) I noticed that the reconstruction does not perform well regarding the correlation in the cross-validation test. Nonetheless, the authors use it to predict the sign of the NAO for which the method seems to be correct about 70 % of the time. I don't really understand the explanation the authors give (p13). It would help if a figure of the Ortega reconstruction and the new reconstruction was shown.*

We will exchange Fig. 6 by the attached figure in which the Ortega reconstruction is included.

On p13, we refer to Fig. S5 to explain the mismatch in sign due to different timings of transitions. There one can see, that there is a mismatch leading to a lower prediction rate of correct signs around the periods characterized by rapid transitions between two different types of NAO phases (like the one happening during the 12th century). Nevertheless, the transitions themselves are resolved correctly in most cases.

*As for the comparison of the present reconstruction with other reconstructions it should be noted that both the reconstruction methodology and the proxy selection will be important. I would suggest that the authors produce a reconstruction from their proxies using a simple multiple regression scheme between the NAO and the proxies. This might help getting an idea of which improvements the network method actually brings.*

If one uses the proxy network used in this study and instrumental NAO data as calibration data this would basically result in a reconstruction similar to the one by Ortega et al. They have chosen the most suitable paleoclimate records in the region to fit to an instrumental NAO time series. The data underlying the Ortega reconstruction have a significant overlap with our selection. Our approach is to use precisely the non-stationary relationship between many paleoclimate archives and the NAO. Thus, any linear regression with this extended data set cannot yield the same results as the network based study, since most of the records do not have a stationary correlation with the NAO. They are thus unsuitable for a linear regression, which relies on a stationary relationship.

*3) Section 3.2: It seems that the similarity is defined from the p-values alone. In my understanding it should be based on a combination of the size of the correlation and the p-value. As a large correlation can be insignificant so can a small p-value be connected to a weak correlation.*

Correlation values can be artificially elevated by large persistence of individual proxies (e.g. in case of prominent low-frequency variations). This effect would be proxy- and site-specific. Therefore, taking the correlation value itself into account is not meaningful.

*The similarity does not seem to take the sign of the correlation into account. From a physical point of view there is a big difference if two point are positively or negatively correlated. So is not a lot of information lost in this process?*

As we argue in the manuscript, different proxy types can indeed lead to different signs and magnitudes of correlations. It might be possible to include the correlation strength (preferably measured by the corresponding p-value as a measure of statistical significance of the existence of a pairwise correlation) into our regression framework, but for simplicity reasons, we preferred to follow a binary approach.

When it comes to the NAO, the absolute values of correlations need to be taken into account in our regression framework (with the number of significant pairwise correlations rather than their directionality as predictors), since the NAO can have opposite yet significant effects in different regions (like higher temperatures in Greenland and lower ones in Fennoscandia) for the same phase. Specifically, in our approach, we are only interested in the existence of a linkage at a given point in time, rather than its specific form.

*4) Introduction, page 2: Networks probably have some advantages in some situations. However, networks were developed for studies of discrete phenomena such as those in sociology. In the study of climate we deal with fields that are continuous in both space and time. It therefore seems backwards to reduce the problem to a network. We must loose information that other methods based on fields take into account. I know that the present paper is not the place for a philosophical discussion but the concern could be mentioned.*

We appreciate this comment. However, one could also argue vice versa, that a network is only using the information which is actually present in the given data set, without consideration of non-sampled regions. The latter would be (at least implicitly) the case in climate field reconstruction methods. In this sense, we consider the spatially discrete network structure as a more honest representation of the sparse data that we have in a paleoclimate context rather than a reduction or a step backwards. We mention this difference in philosophy in line 10 and will elaborate on this in more detail in our revised manuscript.

*Is the A in Eq. 1 used anywhere?*

Yes, for example in Eq. 4. We are confident that using this mathematical symbol, which is standard in complex network theory, will foster an easy access to our manuscript for readers that are familiar with networks. Therefore, we would like to keep both the mathematical and the verbal/intuitive representation of our formalism as parts of the manuscript.

*Caption to Fig. 4 should be improved.*

We will provide a more detailed caption in our revised manuscript.

*Page 7, top: I don't see how the AAFT procedure can be applied to the 4 incomplete proxies. The AAFT includes a Fourier transform.*

Data gaps have been filled by linear interpolation. As we are dealing with regularly sampled data with a very low number of missing data points in the first few centuries only, we are confident that this interpolation does not alter the overall results significantly.

*Fig. 5: The bright areas are not easy to see. By the way: Is CE an accepted standard? It always takes me a while to figure out the direction of the axis.*

We could of course make the bright areas darker, but they are meant to be in contrast to the dark areas, as they mark periods at which the probability for one or the other phase is not high enough to make any conclusive statements. We furthermore think that CE (Common Era) is nowadays a well established time scale without explicit religious connotation, and the direction of the axis is used in most publications on Late Holocene climate dynamics.

*P9,l5: More recent and complete references are Christiansen 2014 (10.1175/JCLI-D-13-00299.1) and Christiansen and Ljungqvist 2017 (10.1002/2016RG000521).*

We thank the reviewer for this suggestion. We will update the list of references accordingly.
* * *
[Figure]

**Fig. 1.**

---

## Author Comment (AC2) · 15 Jun 2017

*This paper presents an interesting statistical methodology to analyze links between different proxy records through a functional network. The authors apply it to a few continental data centered around the east side of the North Atlantic region and covering the last two millennia. Then, they hypothesize that the main network patterns found are related with NAO variations, and use this possibility to try to reconstruct the main signs of NAO variations over the last two millennia. I have found the paper well-written and mainly clear, except at a few occasions where clear definitions were missing I think. The new statistical approach is well explained and sounds promising. It indeed*

[Figure]

*provides an original viewpoint concerning the variations of different records over the last two millennia, the non-stationarity of the links, etc. In that sense, this paper is interesting and deserves to be published.*

We are grateful to the reviewer for this positive overall judgement of our manuscript.

*Nevertheless, I have found the scientific objective of the paper a bit blurry so that it needs to be clarified. For instance, it is not very clear to me why the authors finally jump towards a NAO reconstruction, which is furthermore poorly validated when looking at the few tests they have performed. From my point of view, it would have been nice to better validate the model through the use of pseudo-proxy approach for instance within a climate model, to demonstrate already that we can reconstruct NAO sign variations within a climate model world. I understand that I am here asking possibly a lot, and the paper is already complex, but I should admit that I am not entirely convinced by the approach as it stands. If the authors want to reconstruct the NAO over two millennia, why do not they use classical methods and applied them to their data? This remained unclear to me what the functional network approach brings here.*

As already acknowledged by the reviewer, we make use of the non-stationarity of the relationship between the NAO and the different paleoclimate records. Most classical methods (like linear regression) that we are aware of in turn assume a stationary relationship, which is not a realistic assumption in this case. Thus, our approach tries to extend the knowledge of past NAO variability by relying on exactly the characteristic non-stationary influence of the NAO on many paleoclimate records which is undesirable in applications of other methods.

The same problem also makes it very hard to perform a test using pseudo-proxies. We do not conceptualize the paleoclimate records under study as sole records of temperature + noise but as also sharing additional influences like extreme winter precipitation, connected to the NAO. Thus, one would have to use very elaborate pseudo-proxies, which are also able to capture such effects. While this is indeed a very interesting

question, it clearly surpasses the scope of this research paper. We agree, however, that it would be a subject worth further investigations.

**Specific comments**

*- p. 1, l. 6: "intimately": not sure this intimacy has been really proven. I will use another word here, or just remove it.*

This term can indeed be deleted.

*- p. 1, l. 8-9: "strong co-variability (. . .) as being indicative of a positive phase of the NAO". I think this link needs to be better demonstrated to support such a strong claim.*

Indeed, "co-variability" is enough. We think that the performance of the model, though not perfect, justifies the statement, that certain cross link densities correlate with certain NAO phases and can thus be used as indicators.

*- p. 1, l. 21: the authors cite here the AMO (also called AMV), but they do not discuss it any more afterwards. Why such a focus on the NAO, while the AMV could have played a strong role in past climate variability as well? Please clarify.*

We focus on NAO mainly, because the AMO does not seem to be unambiguously reflected in the spatial co-variability represented by the data. Efforts to employ the same methodology for an AMO reconstruction did not yield any reasonable results.

*- p. 7, l. 18: "|CMtw|". Can you define clearly what the "||" is meaning here.*

This describes the cardinality, the number of members of a set, in this case the number of records in a cluster. We will clarify this in our revised manuscript.

*- P. 8, l. 20: I think it is worth describing in a few sentences what the Louvain algorithm is.*

This is a standard algorithm for identifying cluster structures in network science. We believe that explaining the details of this algorithm beyond just giving a corresponding reference might rather distract the presentation of our work than providing any relevant information to the reader.

*"probably related with NAO": this is a hypothesis. . . Why not the AMV (at low frequency, can play a large role. . .)?*

Of course, AMV can play a large role, too. However, as mentioned above, we have not been able to establish any connection to AMV using our network method. We emphasize that NAO and AMO have distinct spatial patterns, and the available set of proxies used in our study appears more suitable for reconstructing low-frequency variations of the NAO rather than AMV associated with other characteristic spatial co-variability patterns.

*- P. 9, l. 12: why 50-year smooth. Have you tried other smoothing?*

Yes. The results do not change much for larger time windows, for much smaller ones the uncertainty of the pearson correlation becomes too large. We will add a corresponding note to our revised manuscript.

*- P. 9, l.14: How many data in your network and Ortega et al. (2015) reconstruction are in common. It is worth clearly specifying which.*

We will specify this in the table of records in our revised manuscript.

*- Figure 3: the colors and numbers of clusters is unclear. Please be more precise on the methods used to make this figure.*

We will clarify this in our revised manuscript.

*- P. 10, l. 4-9 and Figure 4: I find the interpretation and choice of the slots shown a bit subjective. How do you choose them? How do you identify the main patterns?*

The patterns selected for these features indeed represent a subjective choice and are

just meant to illustrate both the general patterns as well as the limitations when just looking at the networks alone. To make the associated statement more objective is actually the main reason for using the linear model.

*- Figure 5: the legend is not detailed enough to understand how it has been computed.*

The black circles represent the geographic centers of each cluster of proxies. Red links indicate pairs of spatial clusters for which the temporal variations of the corresponding CLDs correlate positively with the NAO phase, while the blue ones indicate negative correlations. The color and width of the links are determined by the sign and values of the regression coefficients of the linear model as given in Tab. S2. We will clarify this in our revised manuscript.

*- P. 12, l. 1-8: I find the numbers for the validation a bit worrying. Even the 68 and 71% are small when considering that in fact you have always 50% chance of being in one phase or another. The authors claim "our results have a certain value". Can you develop a statistical test to be more convincing? Value for what?*

As we already argued in the manuscript, most of the mismatch is due to a difference in transition times between positive and negative NAO phases in the Ortega reconstruction and our model (a problem, which has been solved by wiggle matching in previous reconstructions). Thus far, we are not aware of any alternative framework yielding more convincing results.

*- P. 13, l. 1: How do you prove the claim from this first sentence? This is not clear to my eye when looking at Fig. S5. You should better support this interesting claim.*

In Fig. 5 one can see that at several occasions there is a mismatch during a transition between NAO phases, but our reconstruction seems to be lagged w.r.t. the Ortega reconstruction (e.g. around 1200, 1500 and 1850 CE).

*- P. 13, l. 2: "period of strong, persistent positive or negative NAO": can you provide a clear definition of what this is?*

Here we mean, that the Ortega reconstruction is far from 0 and consistently in one phase during a certain time window, unlike e.g. 1400 CE. We agree that this sentence should be rephrased, stating that a good consistency between the Ortega reconstruction and our model is commonly accompanied during periods with persistent NAO phases.

*- P. 13, l. 15: "degree of belief": can you provide a definition for this?*

This is the probability, determined from the MCMC ensemble, that the NAO is in a specific phase. This can be derived immediately from the linear model. Through using a MCMC based estimator we achieve uncertainty estimates of our model parameters. We then draw a large number of realizations of the model parameters and calculate an estimate of the NAO index based on the observed CLDs. The percentage lower (higher) than 0 is interpreted as the probability, that the NAO was in a certain phase. We will state this more explicitly in our revised manuscript.

*- P. 14, l. 30: "Supplementary Fig. S7": I think that taking 50 years is maybe too short here, and can easily induce artificial non-stationarity just through low frequency.*

We use 50-year windows to generate the network representations and, thus, need to use a similar time scale to demonstrate the non-stationarity of the relationship between the considered proxy records and the NAO. While it is true, that the estimated correlations will exhibit a certain temporal variability due to the short window size, this type of non-stationarity is not the main point here. What we want to demonstrate is the absence of a correlation between the records and the Ortega reconstruction at many times, which we believe cannot be an effect of the windowed analysis alone. This type of non-stationarity (correlation only at certain times) makes a direct, linear regression unsuitable.

*- P. 15, l. 7: I do not understand well what is the final MCMC regression model and how the r2 of 0.58 is computed, can you please clarify?*

The final regression model is the linear model given in eq. 5. The model parameters are estimated using an MCMC approach. The $r^2$ is given by the square of the correlation of the fitted model to the Ortega reconstruction. In any way, we argue, that this number might be due to overfitting of high-frequency variability, as the $r^2$ in the cross-validation is considerably lower.

*- P. 15, l. 14-15: The claim that low-frequency temperature variations are related with solar and volcanism is not so clear in my mind, and on the opposite, there is a large debate on that. See for instance Schurer et al. (2013) or PAGES2K-PMIP3 group (2015). The internal climate variability could have played a large role as well in the last two millennia, even to explain little ice age and medieval climate anomaly.*

This formulation is indeed somewhat misleading. What we wanted to express here is, that while the NAO has an influence on low-frequency temperature variations there are other factors, like solar forcing, volcanism and other modes of internal variability which might be equally or even more important. Thus, a discrepancy between the observed temperature and the temperature expected from a reconstructed NAO phase is not problematic. We will clarify this in our revised manuscript.

*- P. 17, l. 12: "most droughts indeed coincide". In a paper with such advanced statistical tools, I'm surprised to read that. Can you quantify this more precisely, to improve my degree of belief in this claim?*

Out of the 20 drought events there is a tendency towards a positive NAO (P(NAO+) > 0.5) in 17 cases, with 12 of these with P(NAO+) > 0.66. We will add this information in our revised manuscript.

*- P. 19, l. 2: "most likely": is there any statistical test supporting this adverb?*

Here it is not meant to be a statistical term or represent a clear gradation of likelihood like used in the regular IPCC reports. We will clarify this in a revised manuscript.

*- P. 19, l. 14: "Thus, our approach cannot yet be directly applied to the instrumental*

*record as regression target". Why that? Indeed, this would have been nice to further test the method on instrumental record (cf. pseudo-proxy approach from my main comments).*

The main reason is, that thus far we have only been able to perform this analysis for windows of 50 years length. As the instrumental record goes back to 1821 (Vinther et al 2003) this leaves about 4 independent data points for the regression. If one is able to reduce the window size and complexity of the regression model further, one might be able to perform the analysis on the instrumental record, which is highly desirable. We will clarify this in our revised manuscript.

*Bibliography: PAGES 2k–PMIP3 group (2015) Continental-scale temperature variability in PMIP3 simulations and PAGES 2k regional temperature reconstructions over the past millennium. Clim. Past, 11, 1673–1699, 2015 Schurer, A. P., S. F. B. Tett, et G. C. Hegerl, 2014 : Small influence of solar variability on climate over the past millennium. Nature Geoscience, 7 (2), 1–5, doi :10.1038/ngeo2040*

We thank the reviewer for the suggestions and will consider them in a revised manuscript.

---

## Author Comment (AC3) · 15 Jun 2017

*In this paper, the authors use a network approach to investigate climate teleconnections across the North Atlantic region during the Common Era and relate climate in this region to the NAO. The authors take an interesting approach toward utilizing published paleoclimate records for reconstructing a regionally important climate index. I agree with the other Referees that this work should be published with some modifications outlined below. I am in agreement with Referee 1 who suggests that the authors take a more "pedagogical" approach toward describing their methodology. Whenever possible, relating the purpose of equations in words as well, providing definitions for all*

[Figure]

*variables, and including a table of variables that readers can refer back to would help to clarify the approach taken by the authors. This will make their work more accessible and thus their approach will more likely be followed by others in the future.*

We will clarify the general approach and add the requested additional information in our revised manuscript.

*I suggest that the authors include more discussion of the proxies they are including in their analysis, which archives they come from, and what the records are interpreted to show across the interval in question. If the authors are not space limited, I would suggest moving the table of proxies into the main text so that readers can clearly see what records are being used and so the original citations for the records can be included in the main text citations. As this is only 37 records, it does not seem unreasonable to include in the main text. I also suggest a more careful description of the clustering of sites with some information to validate this approach – to show that each proxy does reflect regional temperature to a reasonable degree and can be clustered with other sites in the region.*

We think that the selection of records as well as providing the corresponding details in the Supplementary Material instead of the main manuscript is well justified by the inclusion of most records in the PAGES2k archive, which has reasonable high requirements on the data quality. Also spatial reconstructions from similar data sets justify grouping spatially close records together (see e.g. Werner et al in this special issue). We will clarify this in our revised manuscript.

Regarding the clustering of sites, we have drawn upon recent observation-based records instead the proxies themselves to ensure that the underlying variability of the baseline data reflects temperature directly instead of any site or archive-specific effect. In turn, the fact that the individual archives/proxies are temperature sensitive has been established in the original publications and further discussed in the cited publications of the PAGES 2k working group. Thus, we are confident that this aspect is sufficiently well

documented. This point will be emphasized more clearly in our revised manuscript.

*Some discussion of the uncertainties involved in the proxy records, age uncertainty as well as uncertainty with what each proxy reflects, should also be included. I agree with Referee 1 that to generate space to accommodate clarifications, the section 5.4 could be reduced or removed as it seems overly speculative.*

As discussed in the data section, we only chose annually, or close-to-annually resolved paleoclimate records. They all stem from tree rings, varved lakes and ice cores and can thus, for this time period, be dated very accurately (see also the corresponding discussions in the cited publications of the PAGES 2k working group). Unfortunately, there is no detailed information of proxy uncertainty consistently available for all records. Thus, it is not possible to include this absolutely useful information in our manuscript. See the proxy tables in Werner et al. (same special issue) for an estimate of the uncertainties of some of the varved lake sediment records

As for Section 5.4, we are aware of the fact that our corresponding considerations cannot be proven in the sense of establishing any particular causal linkage between predominant NAO phase and societal development. However, we are confident that it is still illustrative to discuss such observations to highlight the potential impacts of long-term changes in the NAO phase and thus provide potential complementary indications for the qualitative correctness (or at least reasonability) of our reconstruction..

*I also suggest including a plot of the Ortega NAO reconstruction as well as a description of how this was constructed and what records went into it and any potential overlap with the records used in this analysis. Something along the lines of Figure S5 would be useful in the main text to show the comparison between the reconstructions generated here and the Ortega reconstruction.*

We will include a plot of the Ortega NAO in a revised Fig. 6 (see below for a corresponding draft version) and clarify in Tab. S1 which records have been included there.

*Need to explain symbology for site markers – changes from figure to figure, not sure that it means. This is true for both the main text and supplementary figures (e.g. Figure S2)*

In general, we use three types of marker symbols. In Fig. 1, they indicate the type of archive at each location as indicated by the legend. In Fig. 3, we use the same markers for all records in general, but different markers if two clusters are shown in the same color to illustrate, that these clusters are actually distinct. We decided not to use too many colors because this would be even more unclear visually. In all other figures, a circle just marks the center of each group of records. We will explain the markers used in each figure in more detail in our revised manuscript.

*Figure 4 caption needs more description – what do the line thicknesses represent? Why were these time intervals chosen? What controls when points are shown or not shown?*

Here, the line thickness corresponds to the value of the CLD. The time intervals were chosen to illustrate qualitative differences in the large-scale spatio-temporal correlation patterns among our proxies. To find a more objective representation than just this visual representation is the main motivation to investigate the linear model. We will discuss this in more detail in our revised manuscript.

*Elaborate on what is meant by most "informative" clusters and why this is the case (page 9, line 32 – page 10, line 2).*

Most informative means those with the highest regression coefficients in the linear model. We will clarify this in our revised manuscript.

*Are the Deininger et al., (2016) records (mentioned line 25 of page 16) included in the analysis? If not, why not? A diagram of the reconstruction presented here, the Deininger work, and the Ortega reconstruction may be informative.*

They are not included for two reasons. First, they are not temperature proxies and we

wanted to use a set of records which is comparably due to a shared influence of the same climate variable. Second, they do not have nearly-annual resolution, which was one of the main criteria for data selection. We will add a short note regarding the work by Deininger et al. in our revised manuscript.
* * *
[Figure]

[Figure]

**Fig. 1.**

---

## Author Comment (AC4) · 15 Jun 2017

*(1) Expand the "data availability" section to include details on where the input and output data are archived. A Data Citation is needed for the output data (6 below).*

We have submitted the output data as requested to pangeae. It is now available at https://doi.org/10.1594/PANGAEA.875881 and a reference will be added to our revised manuscript. The data availability section of our manuscript will be expanded accordingly.

*(2) Add Data Citations for all of the proxy datasets listed in Table S1.*

We will add missing data citations in our revised manuscript

*(3) For those data not already in a public repository, submit essential metadata along with the time series and include the Data Citation in Table S1.*

We will add all essential metadata in our revised manuscript

*(4) For those records with previous PAGES 2k IDs, include cross references to those IDs in Table S1 (see Table 1 in PAGES 2k Consortium (in press) for PAGES2k IDs). Also, the PAGES2k temperature database version number that was used must be stated in "data availability" and in the title to Table S1.*

We will add the PAGES2k IDs and the database version number in our revised manuscript.

*(5) Explain why records from the PAGES 2k database that meet the criteria for this study appear to have been excluded from this analysis.*

We are not aware of any records in the PAGES2k database which meet the criteria stated in our manuscript and were not included in our analysis.

*(6) Submit the primary outcome of the data analyses to a public repository. This includes: (a) The NAO index reconstruction (Fig S6). (b) The cross-link density time-series in Fig. S4.*

Please see our response to (1). In addition, we will revise the corresponding data file provided in the Supplementary Material accompanying our manuscript.

*(7) Archive the values for the NAO reconstruction in 50-year bins that was used to correlate with the output of this study (last row in Fig. S7; from Ortega et al., 2015)*

Please see our response to (6).

---

## Author Response (AR2)

Dear Prof. Goose,
thank you again for inviting a revised manuscript. We have tried to address all further comments and questions from the reviewers. The modified parts have been marked in the revised manuscript.

**Response to reviewer 1**

This is my second review of this paper. The paper has strongly improved following the first round of review and is now in very good shape, almost ready for publication. It will constitute an interesting input to existing literature on the subject. Nevertheless, I still have a few suggestions for clarifications, which may be mandatory from my point of view, to allow it to be published.

We thank the reviewer for their overall positive response.

1) P. 2, L. 24-26, the AMO potential impacts are claimed to be not "consistently represented in the proxy data", but the reader is left to understand what this is meaning and how such a strong conclusion has been reached. Furthermore, it is a bit weird to have this claim concerning a preliminary result already in the introduction. I would advise to clearly explain in a paragraph or so why the AMV is not consistently represented in the proxy data (which analysis, metrics used…) and to move this in the results section of the analysis

We agree that mentioning this point in the introductory section could be considered a misplacement of information. We have therefore decided to shift the statement mentioning the negative results for AMO/AMV to the end of the conclusions section. Specifically, by applying exactly the same procedure that has been successfully used in our manuscript for extending an existing benchmark NAO reconstruction backwards in time to a similar long-term reconstruction of AMV, we obtained results which did not perform better than a random prediction when comparing the modeled and reference AMV phase. From this observation, we have to tentatively conclude that the considered combination of paleoclimate archives did not allow for modeling the AMV index as target variable using our analysis method. We believe that this finding is important enough (as a cautionary note) to be stated at a prominent place in the manuscript, and have therefore added a corresponding paragraph in the conclusions section. In turn, we think that discussing such negative results in more detail in the results section would not be very helpful.

2) P. 3, L. 32: "stationary manner". I do not get why the authors are doing this claim. Pseudo-proxy approaches are allowing to apply a given statistical methodology to output of climate models, where the true NAO is known, to see notably if there is a kind of stationarity in the reconstruction quality. It is not assuming any stationarity hypothesis. This is mainly a way to test statistical methodology and see within a model world if it works properly when everything is known i.e. reconstructing an index from a few locations, while the dynamical index is known and this, for a long timeframe (last millennium simulations for instance). Can you please further clarify what you have in mind here?

We apologize for any possible misunderstanding regarding this point. In fact, we fully agree that it is possible to have pseudo-proxies also with non-stationary relationships to a target variable, even if they of course have a stationary relationship to the local climate variables.

We argue, that the actual relationship between the NAO index (target variable) and the multiplicity of terrestrial paleoclimate archives considered in our study is potentially complex and in most cases (except for some of the ice core records) not sufficiently well constrained to infer a particular statistical model. In this regard, it would not be sufficient to construct a pseudo-proxy by some relationship to a single model output variable, but actually necessary to model each proxy as a result of the combined action of different variables like summer temperature, precipitation (in our case likely winter extremes) and others. The precise extent to which each variable contributes to a specific archive in a possibly non-stationary manner is in our view not known well enough and would demand sophisticated forward models for all types of paleoclimate archives used in this study. We are not aware of previous studies describing the application of pseudo-proxies that exhibit the corresponding degree of complexity. In our opinion, developing (and subsequently applying) sophisticated pseudo-proxies which are able to reflect this complex relationship (possibly mediated through extreme rainfall, strong storms, etc.) would rather justify a separate study and would expand the present paper beyond reasonable limits. We have clarified this point in our revised manuscript.

3) P. 12, L. 19: Usually the calibration/validation approach is made with an ensemble approach (through random selection of different independent calibration and validation periods), leading to a distribution of r2 that allows to have a better idea on the performance of the statistical model used.

In the paleoclimate literature, splitting the records into two pieces is a standard procedure as well. As we have relatively few independent data points, an ensemble approach (e.g. using block-bootstrapping) as suggested by the reviewer actually yields very similar results. Consequently, we think that splitting the full time interval into two halves provides more insights into the performance of our approach, since the number of records and potentially also the quality of the individual reconstructions decrease as one goes back in time. In turn, this intrinsic difference between different time periods would be widely overlooked in an ensemble approach. Therefore, we prefer to maintain the approach as described in our manuscript.

**Response to reviewer 2**

In my review of the original version I suggested that the authors tried to relate the network properties to more physical properties and in general to take a more pedagogical approach.

I am happy that the authors have followed these suggestions. While the paper is still challenging in all its technical details it is now much more easy to understand the physical reasoning.

As far as I can see the authors have also satisfactorily addressed the comments and suggestions from the other reviewers.

I will therefore now recommend that the paper is accepted for publication.

We thank the reviewer for their overall positive recommendation.

A few minor comments:

P3,l17: The sentence beginning with "Hence, .. ". I don't see how this follow from the previous discussion.

We attempted to clarify this paragraph. It now reads:
"For example, a persistent positive phase of the NAO can enhance winter precipitation in Northern Europe, which in turn has an indirect influence on tree growth during the subsequent summer. The corresponding opposite effect of a negative NAO phase is expected to be much smaller. A similar relationship is expected to be present in Central and Southern Europe, but here increased precipitation is commonly associated with negative NAO phases, while positive NAO phases foster dry conditions and even droughts."

P7, Eq. 2: The x and y's should be normalized for this to give the correlation.

We have added a corresponding remark, that the time series x and y are normalized to keep the notation simple.

P8,l17: No clue ..

Complex network theory provides a great variety of measures for characterizing network properties (clustering coefficients, network transitivity, betweenness centrality, etc.), which have already been utilized in the context of functional climate network analysis in applications to reanalysis data, climate model outputs or dense station grids. All these methods typically request a sufficient number of individual grid points for their appropriate estimation and interpretation, which is not provided in the case of our paleoclimate network. We have further clarified this point in our revised manuscript.

P8, l24: "size" appear twice.

We have removed the second "size".

Fig. 3: As there are only 6 regions the colors could be chosen to be different.

There are indeed more regions highlighted by our cluster analysis. However, some of them do not contain proxy records. In turn, we think that it is important to show the whole domain on which we have applied our cluster analysis for transparency and reproducibility reasons.

P12, l9: Delete "To this end".

We have deleted this opening.